# Malaria outbreak facilitated by engagement in activities near swamps following increased rainfall and limited preventive measures: Oyam District, Uganda

**Maureen Katusiime**[1]*, **Steven Ndugwa Kabwama**[1], **Gerald Rukundo**[1], **Benon Kwesiga**[1], **Lilian Bulage**[1], **Damian Rutazaana**[2], **Alex Riolexus Ario**[1,3], **Julie Harris**[4]

1 Uganda Public Health Fellowship Program, Ministry of Health, Kampala, Uganda, 2 National Malaria Control Division, Ministry of Health, Kampala, Uganda, 3 Ministry of Health, Kampala, Uganda, 4 US Centers for Disease Control and Prevention, Kampala, Uganda

* mkatusiime@musph.ac.ug

**Data Availability Statement:** The datasets upon which the findings are based belong to the Uganda

## Abstract

In April 2019, the District Health Office of Oyam District, Uganda reported an upsurge in malaria cases exceeding expected epidemic thresholds, requiring outbreak response. We investigated the scope of outbreak and identified exposures for transmission to inform control measures. A confirmed case was a positive malaria rapid diagnostic test or malaria microscopy from 1 January—30 June 2019 in a resident or visitor of Acaba Sub-county, Oyam District. We reviewed medical records at health facilities to get case-patients. We conducted entomological and environmental assessments to determine vector density, and identify aquatic *Anopheles* habitats, conducted a case-control study to determine exposures associated with illness. Of 9,235 case-patients (AR = 33%), females (AR = 38%) were more affected than males (AR = 20%) (p<0.001). Children <18 years were more affected (AR = 37%) than adults (p<0.001). Among 83 case-patients and 83 asymptomatic controls, 65 (78%) case-patients and 33 (40%) controls engaged in activities <500m from a swamp (OR$_{MH}$ = 12, 95%CI 3.6–38); 18 (22%) case-patients and four (5%) controls lived <500m from rice irrigation sites (OR$_{MH}$ = 8.2, 95%CI 1.8–36); and 23 (28%) case-patients and four (5%) controls had water pools <100m from household for 3–5 days after rainfall (OR$_{MH}$ = 7.3, 95%CI 2.2–25). Twenty three (28%) case-patients and four (5%) controls did not sleep under bed nets the previous night (OR$_{MH}$ = 20, 95%CI 2.7–149); 68 (82%) case-patients and 43(52%) controls did not wear long-sleeved clothes during evenings (OR$_{MH}$ = 9.3, 95% CI 2.8–31). Indoor resting vector density was 4.7 female mosquitoes/household/night. All *Anopheles* aquatic habitats had *Anopheles* larvae. Weekly rainfall in 2019 was heavier (6.0 ±7.2mm) than same period in 2018 (1.8±1.8mm) (p = 0.006). This outbreak was facilitated by *Anopheles* aquatic habitats near homes created by human activities, following increased rainfall compounded by inadequate use of individual preventive measures. We recommended awareness on use of insecticide-treated bed nets, protective clothing, and avoiding creation of *Anopheles* aquatic habitats.

Public Health Fellowship Pro-gram. For confidentiality reasons, the datasets are not publicly available. However, the data sets can be availed upon reasonable request to jnamagulu@musph.ac.ug and with permission from the Uganda Public Health Fellowship Program.

**Funding:** This Project was funded by the President Malaria Initiative (PMI) and the Cooperative Agreement for Provision of Comprehensive HIV/ AIDS services and Developing National Capacity to manage HIV/AIDS Programs in the Republic of Uganda under the President's Emergency Plan for AIDS Relief (PEPFAR) (Cooperative Agreement number U2GGH001353-04) through the United States Centers for Disease Control and Prevention (US CDC) to the Uganda Public Health Fellowship Program (PHFP)(recipient: MK), Ministry of Health (MoH) through Makerere University School of Public Health (MakSPH). The funder had no role in the design of the study, collection of data, analysis or decision to publish the work. Its contents are solely the responsibility of the authors and do not necessarily represent the official views of the PMI, PEPFAR, US CDC, MakSPH, PHFP or the MoH.

**Competing interests:** The authors have declared that no competing interests exist.

**Abbreviations:** AR, Attack rate; CDC, Centers for Disease Control and Prevention; CI, confidence interval; DHIS2, District Health Information System (DHIS2); CFR, case fatality rate; IRD, indoor resting density; LLIN, long lasting insect side treated nets; ICCM, integrated community case management; IRS, indoor residual spraying; mRDT, malaria rapid diagnostic test; PMI, President's Malaria Initiative; PSC, pyrethrum spray catches; SD, standard deviation; UMOH, Uganda Ministry of Health; $OR_{CLR}$, odds ratio; $OR_{MH}$, Mantel Haenszel odds ratio.

## Introduction

An estimated 229 million cases and 409,000 malaria-related deaths occurred in 2019, with the African region contributing the vast majority [1]. Uganda is a malaria-endemic country with active transmission in 99% of the country. It ranks third in terms of malaria burden worldwide [1]. While all persons in Uganda are at risk of contracting malaria, children under five years of age and pregnant women are the most vulnerable. In Uganda, malaria accounts for 30–50% of outpatient visits at health facilities, 15–20% of all hospital admissions, and up to 20% of all hospital deaths, of which approximately one-quarter occur among children under five years of age [2].

Despite having multiple interventions to combat malaria, such as mass and routine distribution of LLINs through campaigns, schools, antenatal and immunization clinics, IRS and a national test, treat and track policy [3–5], Uganda has long suffered outbreaks of malaria [6]. These have been linked to climatic and environmental factors (temperature, rainfall and humidity),which provide favorable weather conditions for both vectors and parasite reproduction and survival, leading to a high malaria transmission intensity [7–9]. In addition, the Uganda Malaria Reduction and Elimination Strategic Plan 2021–2025 highlights that the country still has inefficient supply chain management of malaria commodities which compromises the appropriate case management, inadequate epidemic prevention, preparedness and response as strategic challenges to malaria control in Uganda [10]. Moreover, having an adequate malaria surveillance systems is compromised by poor data quality, incompleteness and inability to analyze [11] and translate to meaningful metrics of malaria burden [12] to detect epidemics and respond in a timely manner among other malaria control challenges faced in Uganda [7, 13].

In April 2019, the District Health Office in Oyam District reported an upsurge in the number of malaria cases in the district since January 2019, particularly affecting three sub-counties (Acaba, Iceme, and Otwal). While the high season for malaria is normally in June-July, the upsurge during this time period was unexpected. Normal channel graphs (graph showing an upper and lower limit for expected case counts) for malaria cases constructed with data from the District Health Information System (DHIS2) showed that malaria cases had exceeded action thresholds, with more than 1,900 cases reported per week in Acaba Sub-county, Oyam District compared to an average of 1,390 reported in the normal channel upper limit (75[th] percentile of cases over the last five years) (Fig 1). Acaba Sub-county was the most affected of the sub-counties in the district. In July 2019, we conducted an investigation in Acaba Sub-county to determine the scope of the outbreak, identify factors associated with increased transmission and infection, and recommend evidence-based control measures to this outbreak.

## Materials and methods

### Ethics statement

This investigation was in response to a public health emergency and was, therefore, determined to be non-research. The MoH gave the directive and approval to investigate this outbreak. The CDC Center for Global Health determined that this activity was not human subject research, and its primary intent was public health practice or a disease control activity (specifically, epidemic or endemic disease control activity). Verbal informed consent in the local language was sought from respondents or care-takers of diseased children. They were informed that their participation was voluntary and their refusal would not result in any negative consequences. To protect the confidentiality of the respondents, each was assigned a unique identifier which was used instead of their names.

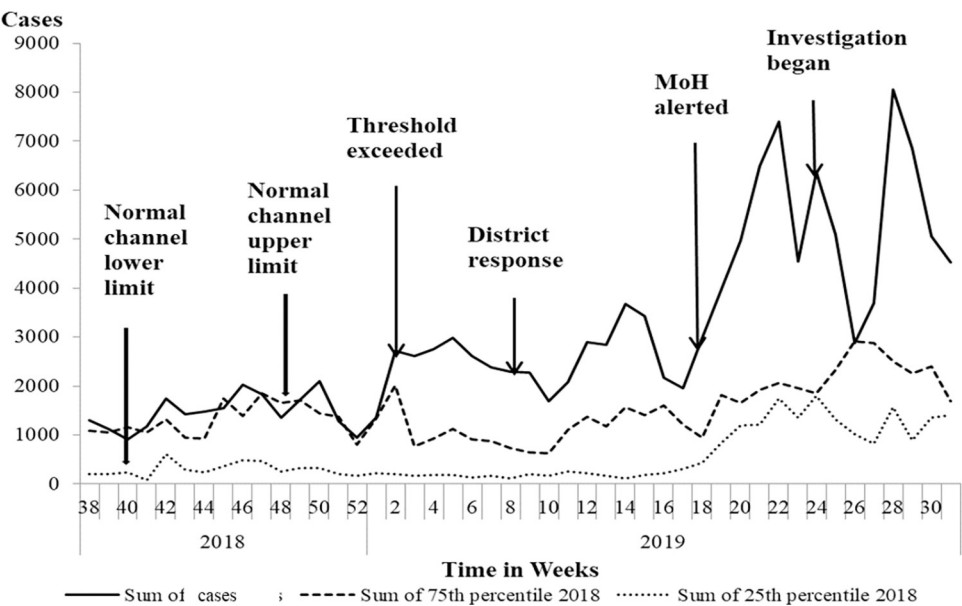

**Fig 1. Normal channel graph showing lower limit, upper limit and malaria cases in Oyam District.**

**Outbreak area.** This outbreak investigation was conducted in Oyam District (population: 426,200) located in Northern Uganda. It has a wet season lasting from April to mid-November and is generally warm and humid, with average monthly rainfall between 12.7mm to 116.8mm, while the dry season, lasting from mid-November to March, is hot [14]. During the dry season, the daily average high temperature ranges from 18.9˚C to 35.0˚C, while in the wet season, it ranges from 18.3˚C to 28.8˚C [14]. Administratively, the district has 12 sub-counties, including Acaba. The main economic activity in Oyam District is subsistence agriculture where farmers grow mainly cotton, cassava, sweet potatoes, millet, pineapples, and bananas. Livestock kept in the district includes cattle, goats, sheep, pigs, chicken and rabbits [15]. Acaba sub-county has four government health centres (HC): one HC II and three HC IIIs.

## Case definition and finding

We defined a confirmed case as a positive of histidine-rich protein II malaria rapid diagnostic test (mRDT) or malaria microscopy from 1 January—30 June 2019 in a resident of or visitor to Acaba Sub-county, Oyam District. We line-listed malaria cases by reviewing outpatient records from all four government health facilities in Acaba Sub-county. We extracted data from the outpatient registers, including age, sex, diagnostic test done and test result, test result date, village, parish, and sub-county of residence for the period January to June 2019. Malaria cases diagnosed by community health workers were captured from the attached health facility.

## Descriptive epidemiology

We constructed malaria normal channels using data for malaria cases confirmed by microscopy and malaria rapid diagnostic test (mRDT) and abstracted from the District Health Information Software 2 (DHIS2). A malaria normal channel is a line graph of 25th and 75th percentiles of malaria cases for the previous five years; it has a lower and an upper limit beyond which it indicates an outbreak of malaria [16]. We disaggregated malaria normal channels by sub-county to reveal those most affected. We calculated attack rates using sub-county specific cases as numerator and sub-county populations as the denominator.

Line lists were obtained from reviewing outpatient medical records at four government health facilities in Acaba. We performed a descriptive analysis of data in the line list, describing case-patients by person, place, and time. We computed attack rates by age, sex, parish, and village of residence based on the Uganda Population and Housing Census data from 2014, projected to 2019 [17]. We constructed an epidemic curve to show the distribution of cases over time from 1 January to 30 June 2019. We also drew a spot map of the sub-county indicating the most affected villages.

## Environmental assessment

Following the descriptive epidemiology, we moved through the most affected villages of Acaba Sub-county (Lelateng and Alao B). We searched for environmental and human factors that may have catalyzed the upsurge in malaria cases during the period. We searched for potential mosquito *Anopheles* aquatic habitats such as stagnant water pools located near affected neighborhoods. We searched for human activity such as construction sites, sand/soil mining sites, brick-making, and new agricultural sites. We identified and defined *Anopheles* aquatic habitats using definitions and methods previously described in studies conducted in Gambia and Uganda [18]. A swamp was defined as a large water body containing vegetation and tall papyrus; puddle as a small natural water-filled depression; pool as a large man-made depression holding water [18]. A pit was defined as a depression made after extracting sand and making bricks; a container as a discarded plastic or metallic waste; rice field as flooded area used to grow rice [18].

## Rainfall

We obtained daily rainfall data for January-July 2018 and January-July 2019 for Oyam District [14]. The rainfall data were aggregated and averaged weekly. We used a t-test to determine the significance of the difference in total rainfall between 2018 and 2019.

## Entomological assessment

**Adult mosquito collection.** We collected adult mosquitoes using pyrethrum spray catches (PSC) in Lelateng and Alao B villages for a period of 4 days. From each village, 20 houses were randomly selected; five were sampled each day. We obtained verbal consent from the household heads and explained the procedures to them. Between 6:00am and 10:00am, we laid white groundsheets inside the houses and sprayed mosquitoes down using an aerosol pyrethrum insecticide called "KILIT Insectcide". KILIT contains d-Tetramethrin 0.135% w/w, d-Allethrin 0.06% w/w and Cypermethrin 0.46% w/w. Mosquitoes were picked using forceps and placed in petri dishes. We identified the collected mosquitoes using an identification key and morphological features on their legs, wings, and palps [19]. We observed the abdominal status of mosquitoes to identify whether or not they had fed the previous night. We dissected unfed female mosquitoes and examined their dried ovaries and determined the parity rate and approximate age of the vectors. Mosquito age was calculated using the following formula [20]:

$$\frac{1}{\log \ell^P}$$

where $\ell$ is the natural logarithm of the constant 2.7183 and $P$ is the probability of the vector surviving for 1 day. $P$ is calculated from the following formula:

$$P = \text{gc}\sqrt{\text{PR}}$$

where gc is the *Anopheles* gonotrophic cycle: the time (in days) needed by a female mosquito to complete the processing of egg development in the ovaries from the blood meal to the time when the fully developed eggs are laid [21]. In this study, gc was estimated at 3 days based on previous research [21]. PR is the proportion of mosquitoes that are parous, calculated as the number of parous females multiplied by 100 and divided by the total number of females dissected [20, 22].

We calculated indoor resting density (IRD) for vectors using the following formula [23].

$$\text{IRD} = \frac{\text{number of mosquitoes captured indoors/number of households}}{\text{number of nights}}$$

**Mosquito larval assessment.** Upon viewing larvae in habitats, we identified the *Anopheles* aquatic habitats around sampled households. We used larval scoops to collect and search for mosquito larvae. We used a standard 350 ml dipper to collect water from stagnant water sources, and took three dips per square meter of stagnant water surface area [24, 25]. We determined larval density by taking the average number of mosquito larvae from the total dips taken at a specific habitat [26, 27].

We further computed larval density in *Anopheles* aquatic habitats around sampled households using the household and container indices using the formulae below [23, 26, 27].

$$\text{Household index} = \left[\frac{\text{Number of houses infested x 100}}{\text{Number of houses inspected}}\right]$$

$$\text{Container index} = \left[\frac{\text{water-holding containers found positive with larvae}}{\text{Total containers searched}}\right] \text{x 100}$$

A container was considered positive if it contained at least one larva [27]. We computed the house index to understand the infestation levels and the container index to know the proportion of water-holding containers that are positive with larvae [26]

## Hypothesis generation

We interviewed 20 conveniently-sampled, confirmed case-persons residing in Lelateng and Alao B villages using a standardized semi-structured questionnaire in order to develop feasible working hypothesis on possible contributors to the malaria outbreak. We asked about the ownership and use of long-lasting insecticide-treated nets (LLIN), time of going to sleep, and reported malaria cases in households and neighborhoods. We also observed the surroundings of the homesteads for the presence of containers with stagnant water, and proximity to potential or active *Anopheles* aquatic habitats.

## Case-control study

We conducted a 1:1 matched case-control study in the most affected parish to test our generated hypotheses. We calculated the sample size using StatCalc in EpiInfo 7, assuming a power of 80%, two-sided confidence level of 95%, a case to control ratio of 1:1, and with 67% of cases and 43% of controls exposed. We considered that living within 3 km of a river or swamp was a significant risk factor for contracting malaria with an odds ratio (OR) of 2.7 [28]. Using these assumptions, we calculated the minimum required sample size to be 66 cases and 66 controls. With a 20% non-response rate, our sample size was expanded to 83 cases and 83 controls [29].

We selected cases randomly from our line list and identified one village control per case. If a household had more than one case, only the case-patient with earliest date of onset was

enrolled. We selected cases and controls by village proportionately to the village-specific attack rates. We selected households in which no member had had fever during the previous 3 months and selected a control closest in age to the case-patient (+/- 10 years). We conducted interviews for both the case-patients and controls using a structured questionnaire for the effective exposure period (2 weeks before the matched case's illness onset), asking about ownership and usage of an LLIN, living <500m from a swamp, engagement in commercial ventures <500m from a swamp, engagement in late evening outdoor activities, having water pools <100m from household for 3–5 days after rainfall, wearing long clothes during evening hours, not drawing curtains on doors, and entering bed to sleep after 9pm. Data collected on the respondents' demographics included age, sex, occupation, and employment status.

We developed case-control sets and used EpiInfo 7.2 to conduct descriptive analysis and 2x2 tables to cross-tabulate outcome variables with exposures. If case-patients or controls reported multiple activities near swamps, we chose their exposure for analysis to be the closest distance from the swamp. We stratified the outcome variables with other exposures to obtain Mantel-Haenszel odds ratios with 95% confidence intervals, this provided the magnitude of association. We used the chi-square test to establish differences among categorical variables and groups. We set a p value of <0.05 as sufficient evidence to reject the null hypothesis of no difference.

## Results

### Descriptive epidemiology

The malaria normal channel showed that, starting from epidemiological weeks 2 of 2019 onwards, malaria cases exceeded the action threshold in Oyam District (Fig 1). Disaggregation of malaria case-patients by sub-county showed Acaba Sub-county as the most affected (AR = 40%), followed by Oyam Town Council (AR = 38%) (Fig 2). The following analyses are focused on Acaba Sub-county.

We identified 9,235 case-patients (AR = 33%) who visited four health facilities in Acaba Sub-county from 1 January-30 June 2019; one case-patient died. Attack rates among the six parishes ranged from 20% to 48% (Fig 2). The median case-patient age was 20 years (IQR: 6–21 years) and the mean age was 26 ±18.1. Females (AR = 38%) were more affected than males (AR = 20%) (p<0.001). Children <5 years (AR = 38%) and persons aged 5–17 years (AR = 38%) were both more affected than persons ≥18 years of age (AR = 27%) (p<0.001).

The epidemic curve showed a mostly flat curve of malaria cases from January through April (Fig 3). Rains began during early March 2019 and had increased by early April. Malaria cases increased from late April and May, peaking in mid-May. Oyam District began reporting ACT stockouts in mid-April, and reporting ACT and RDT stockouts at two out of four facilities in Acaba Sub county during the latter part of May. The rains from January to June 2019 were heavier (6±7.2mm) than during the same period in 2018 (1.8±1.8mm) (p = 0.006).

### Environmental assessment findings during a malaria outbreak, Acaba Sub-county, January—June 2019

Much of the land in Acaba Sub-county is swampy, and during our visit we observed pools of stagnant water in many areas harboring mosquito larvae (Fig 4). Following increased rainfall in early April 2019, communities encroached on the swamps to carry out yam and maize farming, animal grazing, brick making (excavating soil to make bricks, which creates pits), and creation of large-scale rice fields, all of which created the potential breeding environments for mosquitoes. Households were also observed to have uncovered water-holding containers outside the house, and many sand pits with stagnant water were observed in the community.

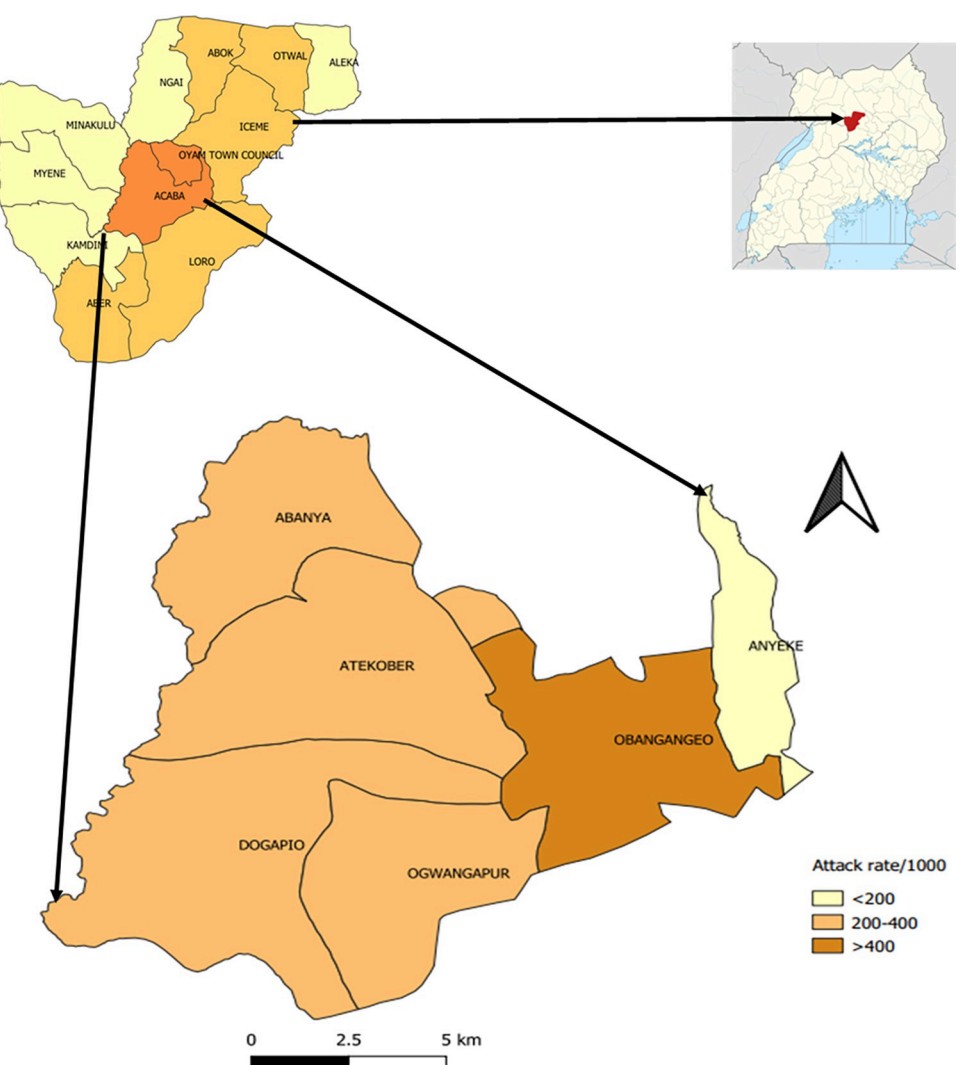

**Fig 2. Map showing attack rates per 1,000 population by parish in Acaba Sub-county, Oyam District.** Fig 2 was drawn using qgis version 3.2 and subcounty shape files from Uganda Bureau of Statistics (UBOS). The base layer for the map was obtained from within qgis version 3.2. qgis is an open-source software or open license. We linked subcounty shape files from UBOS to the base layer within qgis; How to link shape files using vector files in qgis: https://docs.qgis.org/3.16/en/docs/user_manual/working_with_vector_tiles/vector_tiles_properties.html.

### Entomological assessment findings

Among 1,151 mosquitoes identified from PSC, 955 (83%) were malaria vectors, including 902 (94%) *An. gambiae sensu lato (s.l)* (80% female) and 53 (6%) *An. funestus* (72% female). Overall combined 192 (20%) *Anopheles* male mosquitoes were identified. The average IRD for *An. gambiae s.l* was 4.5 female mosquitoes /household/night while that for *An. funestus* was 0.2.

Of 720 female *An. gambiae s.l* females, 466 (65%) were fed. Among the 254 unfed, 219 (86%) were nulliparous, and 35 (14%) were parous. Of the 38 female *An. funestus*, 17 (45%) were fed, and all 21 unfed (100%) were nulliparous. We found a parity rate of 14% and longevity of 2.1 days.

A total of 59 *Anopheles* aquatic habitats were sampled and investigated <100m from households; sand pits [3], swamp edges [1], brick pits [1], puddles [2] and 52 containers. All the

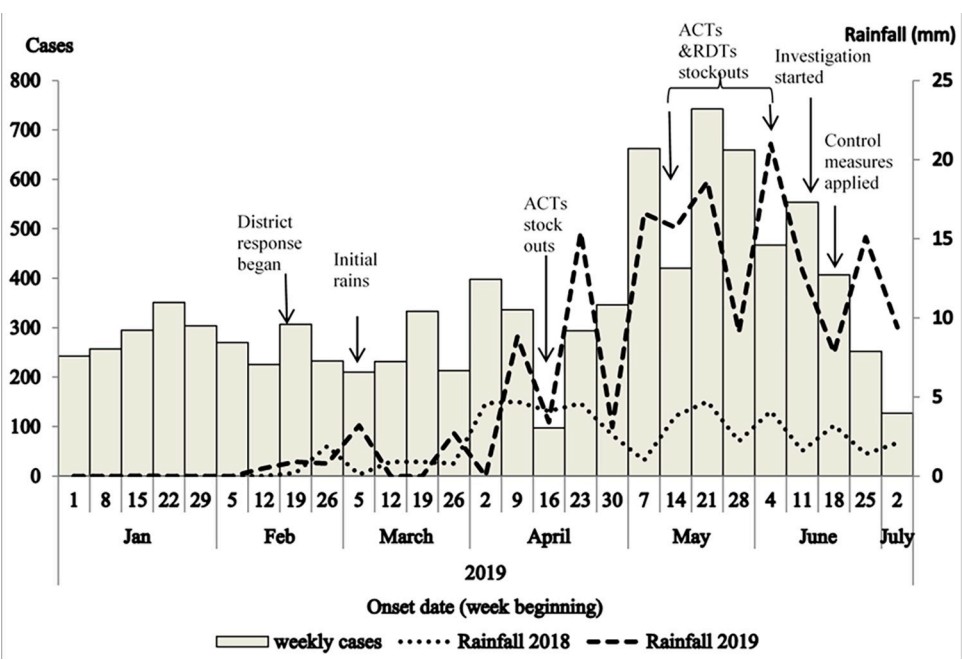

**Fig 3. Week of symptom onset of case-patients and rainfall amount during a malaria outbreak: Acaba Sub-county, Oyam District.**

*Anopheles* aquatic habitats had *Anopheles* larvae; sand pits (25/350mls) and brick pits (16/350mls) had the highest average larvae concentrations per scoop.

We found a household index of 5.4% and a container index of 21%.

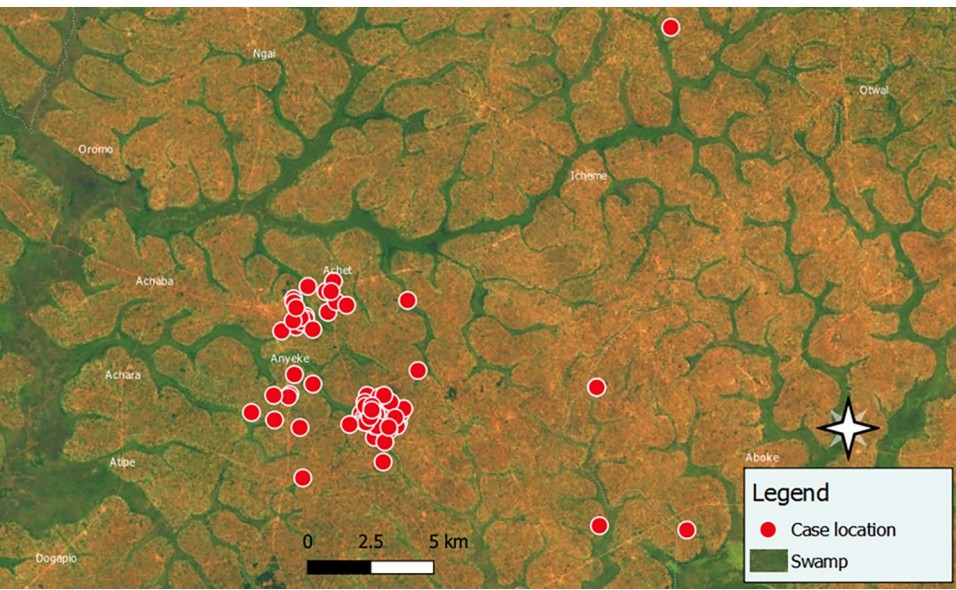

**Fig 4. Distribution of cases and swamp location during a malaria outbreak: Acaba sub-county Oyam District.** The baselayer for the map contains information from OpenStreetMap and OpenStreetMap Foundation, which is made available under the Open Database License. It also contains information from qgis, an open-source software or open license. Within qgis, we linked subcounty shape files from UBOS to the base layer. Vector files: https://docs.qgis.org/3.16/en/docs/user_manual/working_with_vector_tiles/vector_tiles_properties.html.

## Hypothesis generation

Of the 20 case-patients interviewed, 14 (70%) carried out activities within 100m of a swamp, and 13 (65%) entered bed after 9pm. We hypothesized that having outdoor activities within 500m of a swamp was associated with malaria infection. These activities included cooking outside the house late in the evening (most households had open cooking spaces detached from the household), family gathering around a fire after 7 pm, attending to roadside evening markets, brick making, road construction, crop cultivation, and animal rearing within and near swamps, among others.

## Case-control study

The 83 case-patients and 83 controls were similar to each other by sex, age, employment status and occupation (Table 1).

All case-patients and controls engaged in daily activities near a swamp. Factors that increased odds of malaria included engaging in activities <500m from a swamp, living <500m from irrigated rice field, and having stagnant water pools <100m from household for 3–5 days

**Table 1. Characteristics of cases and controls in case control study, Oyam District, January-June 2019.**

| Variable | Cases (n = 83) | Controls (n = 83) | Cases % | Controls % |
|---|---|---|---|---|
| **Sex** | | | | |
| Male | 44 | 48 | 53 | 57 |
| Female | 39 | 35 | 47 | 43 |
| **Age group** | | | | |
| <5 years | 4 | 1 | 5 | 1 |
| 5–17 years | 25 | 31 | 30 | 37 |
| ≥18 years | 54 | 51 | 65 | 62 |
| **Employment Status** | | | | |
| Employed | 51 | 48 | 62 | 58 |
| Not employed | 32 | 35 | 38 | 42 |
| **Occupation** | | | | |
| Farmer | 48 | 47 | 58 | 57 |
| Student/Pupil | 27 | 32 | 32 | 38 |
| Others | 8 | 4 | 10 | 5 |
| **Activities near swamp**[a] | | | | |
| **Animal Farming** | | | | |
| Yes | 33 | 18 | 39 | 22 |
| No | 50 | 65 | 60 | 78 |
| **Crop cultivation** | | | | |
| Yes | 58 | 27 | 69 | 33 |
| No | 25 | 56 | 30 | 67 |
| **Fishing** | | | | |
| Yes | 3 | 5 | 4 | 5 |
| No | 80 | 78 | 96 | 95 |
| **Brick making** | | | | |
| Yes | 9 | 9 | 11 | 11 |
| No | 74 | 74 | 89 | 89 |

*Mean and standard deviation; Mean age = 26 ±18.1 for case-patients and 27 ±18.2 for controls

[a]Activities near swamp included: Animal farming, crop cultivation, Fishing and Brick making among others

after rainfall. Protective factors included sleeping under LLINs and wearing long-sleeved clothes during evening hours (Table 2).

## Discussion

Cases of malaria in this outbreak were associated the presence of mosquito *Anopheles* aquatic habitats, created by activities near and within swamps, within a few meters of the home. This was compounded by waterlogged soil and stagnant water in discarded containers around homes. Significantly heavier rains during 2019, compared with 2018, likely contributed to increases in mosquito *Anopheles* aquatic habitats and the upsurge identified during this outbreak. Notably, 20% of mosquitoes captured in homes were male, potentially representing that their aquatic habitats were close to households. Failure to adhere to individual protective activities provided additional opportunities for mosquito bites and malaria transmission.

The Ugandan population is increasingly using swampy land for both formal and informal commercial activities [30]. These include: agriculture [31], rice irrigation, livestock grazing [32], fishing [33], and brickmaking [30], among others. Moreover, the recent human encroachment on swamps has resulted into environmental degradation, which changes the ecological balance and context in ways that can promote malaria. An example is having elevated temperatures that facilitate mosquito breeding and increase the rate of development of the parasite inside the mosquito [30, 34]. People who have settled in or near swamps may also face frequent flooding, which can compound malaria outbreaks through the expansion of mosquito *Anopheles* aquatic habitats [30]. Our findings are consistent with other studies that have found that human activities in, around, or near swamps were associated with malaria outbreaks in Uganda [23], Tanzania [35], Kenya [36], and West Africa [37].

Rainfall may also have played a role in this outbreak. People always get malaria because their activities and lifestyle put them at risk [7, 8, 10]. We found that activities near and within swamps such as animal farming, crop cultivation, rice growing, soil excavation, brick making were implemented following the unusually high levels of rain when compared to the previous year. However, it is not clear if these activities actually changed or if it was the heavier rainfall in 2019 that made the activities that had always been done high-risk, even without changes in activities. Although it is possible that *Anopheles* aquatic habitats were formed during the heavy rainfall period which may have allowed for unusual increase in the vector population. Supporting this theory, the household and container indices were high in affected areas, and a substantial proportion of mosquitoes found in the home were male. This suggests that *Anopheles* aquatic habitats may have been very close to or inside homes, as male mosquitoes are weaker fliers than females [26], do not need human blood and typically avoid human contact [38]. However, the case threshold indicating an outbreak was crossed before the rains started, suggesting that, while the rainfall may have compounded the outbreak, other factors contributed to its initiation. These might include the inadequate epidemic preparedness and response and the inefficient supply chain management of malaria commodities. These have been highlighted as strategic challenges towards malaria control in the Uganda Malaria Reduction and Elimination Strategic Plan 2021–2025 [10]. To support this further, we found that both malaria tests and malaria treatment faced stock-outs during our investigation. This may have compounded the problem as they forced people to either go undiagnosed or find alternative, less effective treatments. This highlights the multifactorial nature of malaria outbreaks and the challenges in addressing them comprehensively.

Inadequate LLIN use in the context of a highly vulnerable environment likely contributed to this outbreak as well as to the normal endemic burden of malaria. As of 2019, 83% of households in Uganda owned at least one insecticide treated net (ITNs); 54% of households had at

**Table 2. Exposures associated with malaria infection during an outbreak: Oyam District, Uganda, January-June 2019.**

| Exposure | Exposed (n) | | Exposed (%) | | OR M-H | 95% CI | P-value |
|---|---|---|---|---|---|---|---|
| | Case (n = 83) | Control (n = 83) | Case | Control | | | |
| **Vector breeding or harborage** | | | | | | | |
| **Overgrown bushes around home** | | | | | | | |
| Yes | 42 | 12 | 51 | 14 | 11* | 3.4–36 | P<0.001 |
| No | 41 | 71 | 49 | 86 | ref | | |
| **Distance of human activity from swamp** | | | | | | | |
| <100m | 28 | 8 | 34 | 10 | 12* | 3.3–47 | P<0.001 |
| 101-500m | 39 | 39 | 47 | 47 | 5.4* | 1.6–19 | P = 0.008 |
| >500m | 16 | 36 | 19 | 43 | ref | | |
| **HH within <500m from swamp** | | | | | | | |
| Yes | 66 | 33 | 80 | 40 | 12* | 3.6–38 | P<0.001 |
| No | 17 | 50 | 20 | 60 | ref | | |
| **Waterlogging around HH for 3–5 days after rainfall** | | | | | | | |
| Yes | 23 | 4 | 28 | 5 | 7.3* | 2.2–25 | P<0.001 |
| No | 60 | 79 | 72 | 95 | ref | | |
| **Distance of HH from irrigated rice field** | | | | | | | |
| <500m | 18 | 4 | 22 | 5 | 8.2* | 1.8–36 | P = 0.005 |
| 500m to 1km | 7 | 7 | 8 | 8 | 1.2 | 0.4–3.8 | P = 0.66 |
| >1km | 58 | 72 | 70 | 87 | ref | | |
| **Stagnant water in discarded containers around household** | | | | | | | |
| Yes | 16 | 6 | 19 | 7 | 3.5* | 1.2–11 | P<0.001 |
| No | 67 | 77 | 81 | 93 | ref | | |
| **Distance of HH from soil quarry** | | | | | | | |
| <500m | 6 | 2 | 7 | 2 | 5.9 | 0.9–40 | P = 0.07 |
| 500m to 1km | 7 | 3 | 8 | 4 | 4.5 | 0.8–24 | P = -0.08 |
| >1km | 70 | 78 | 84 | 93 | ref | | |
| **Distance of HH from road construction** | | | | | | | |
| <500m | 17 | 21 | 20 | 25 | 0.3 | 0.1–1.7 | P = 0.18 |
| 500m to 1km | 0 | 3 | 0 | 4 | 0 | indefinite | P = 0.96 |
| >1km | 66 | 59 | 80 | 71 | ref | | |
| **Accessibility of humans to feeding vectors** | | | | | | | |
| **Didn't sleep under LLIN last night** | | | | | | | |
| No | 23 | 4 | 28 | 5 | 20 | 2.7–149 | P<0.001 |
| Yes | 60 | 79 | 72 | 95 | ref | | |
| **Didn't wear long clothes in evening** | | | | | | | |
| No | 68 | 43 | 82 | 52 | 9.3* | 2.8–31 | P<0.001 |
| Yes | 15 | 40 | 18 | 48 | ref | | |
| Had late evening outdoor activities* | | | | | | | |
| Yes | 36 | 10 | 43 | 12 | 5.3* | 2.2–13 | P<0.001 |
| No | 47 | 73 | 57 | 88 | ref | | |
| **Household size (>5 vs. ≤5)** | | | | | | | |
| >5 members | 24 | 13 | 29 | 16 | 1.6 | 0.8–3.5 | P = 0.07 |
| ≤5 members | 59 | 70 | 71 | 84 | ref | | |
| **HH lacked curtains on doors** | | | | | | | |
| Yes | 59 | 52 | 71 | 63 | 1.6 | 0.8–3.5 | P = 1.24 |
| No | 24 | 31 | 29 | 37 | ref | | |
| **Kitchen not attached to main HH** | | | | | | | |

*(Continued)*

**Table 2.** (Continued)

| Exposure | Exposed (n) | | Exposed (%) | | OR M-H | 95% CI | P-value |
|---|---|---|---|---|---|---|---|
| | Case (n = 83) | Control (n = 83) | Case | Control | | | |
| **Yes** | 80 | 82 | 96 | 98 | 0.03 | 0.0–3.2 | P = 0.25 |
| **No** | 3 | 1 | 4 | 2 | ref | | |

HH = Household, OR$_{MH}$- Mantel Haensel Odds Ratio, CI = Confidence Interval, LLIN = Long Lasting Insecticide Net.

Engagement in late evening outdoor activities include: Family gathering around fire after 7pm, entering bed after 9pm, cooking outside and storytelling outside HH.

*Significant association with malaria infection

least one ITN for every two people (universal coverage of LLIN/ITNs) and 72% had access to ITNs, yet only 59% (of the 72%) used their ITNs [39]. To have the full protective effect of LLIN/ITNs, both coverage of net ownership and use should be above 80% [39]. It is important to note that LLINs would not have protected the persons at risk due to working in swampy areas in this outbreak. However, strengthening of supplementary integrated approaches towards malaria prevention and control [23] coupled with the routine awareness and behavior change communication messages towards malaria could help curb transmission in malaria-endemic areas.

Although malaria is considered a priority epidemic-prone disease in Uganda, response to this outbreak was delayed for eight weeks after the normal channel threshold was crossed [13]. The Uganda Malaria Reduction and Elimination Strategic Plan 2021–2025 highlighted having inadequate use of malaria surveillance data for decision making as a strategic challenge and committed to transforming surveillance into a core intervention to include epidemic prevention, preparedness and response among the strategic shifts of focus in the current implementation [10]. Malaria normal channels are a tool to help districts more quickly identify and respond to unusual case counts for a given region and time period [16]. They are meant to be monitored weekly at all health systems levels, from the health facility to the district and national level [40, 41]. While the aggregated data clearly showed the outbreak, there were critical delays in response between the district and MoH. During this outbreak, the district responded by conducting mass testing and treatment, sensitizing the public on preventive measures. However, it faced stockouts of both malaria tests and malaria treatment, which might have compounded the problem as this forced people to either go undiagnosed or find alternative, less effective treatments. Moreover, the district wrote to the MoH to support the response with emergency supplies of the test kits and drugs which delayed. This outbreak could have been minimized by implementing timely effective control measures. In addition, aggregation of data across many facilities at national level might mask an outbreak, which represents a loss of operational situational awareness for the malaria epidemic preparedness and surveillance system approach. Uganda is moving towards a computerized and web-based surveillance system [42]. Continuing to build the capacity of health facilities to analyze and use their data through provision of computers, training, and mentorship, will be critical to promote early detection of and response to outbreaks, including but not limited to malaria tests and drugs at facilities.

## Limitations

Our data have several possible limitations. First, we did not utilize data generated at private health facilities, as many of them do not use the standard reporting registers or data entry mechanisms. A decline in cases at the apparent start of the investigation also coincided with stock outs of RDTs at health facilities, making it difficult to know the true case burden during

the end of May and the beginning of June. Other factors during the then-dry season may have been at work which we could not ascertain. We did not use transects to assess the 2 km radius of the Anopheles aquatic habitats, instead it was by visual scanning. This was a non-systematic approach which could have led to bias in the results through under or over ascertainment of the Anopheles aquatic sites. Finally, our entomologic investigation was limited: we did not calculate sporozoite rates, entomological inoculation rates and did not conduct deeper interpretation of resting density and mosquito age parameters since these were beyond the scope of this investigation.

## Conclusion and recommendations

This outbreak was facilitated by *Anopheles* aquatic habitats near homes created by human activities. Increased rainfall exceeding the previous year's rainfall likely led to an increased vector population and subsequent increase in malaria cases. This outbreak was also compounded by inadequate malaria control measures to contain the increased vector density and the failure to respond in a timely manner after the normal channel threshold was crossed. We recommended having a mass drug administration of antimalarial treatment to residents in Acaba, increasing community awareness on the use of insecticide-treated bed nets, protective clothing and preventing the creation of *Anopheles* aquatic habitats, including draining sites around homes and support to enhance use of malaria channels at health facilities for prompt outbreak detection and response.

## Acknowledgments

We appreciate Mr. Okello David, the Vector Control Officer, Oyam District for his technical guidance during our entomological assessment. We thank Oyam District Local Government and the Village Health Teams for the guidance in the community during data collection.

## Author Contributions

**Conceptualization:** Maureen Katusiime, Steven Ndugwa Kabwama, Gerald Rukundo.

**Data curation:** Maureen Katusiime.

**Formal analysis:** Maureen Katusiime, Steven Ndugwa Kabwama.

**Funding acquisition:** Julie Harris.

**Investigation:** Maureen Katusiime, Steven Ndugwa Kabwama, Gerald Rukundo.

**Methodology:** Maureen Katusiime, Steven Ndugwa Kabwama, Gerald Rukundo.

**Project administration:** Maureen Katusiime, Benon Kwesiga.

**Resources:** Alex Riolexus Ario, Julie Harris.

**Supervision:** Steven Ndugwa Kabwama, Benon Kwesiga, Lilian Bulage, Alex Riolexus Ario, Julie Harris.

**Validation:** Damian Rutazaana, Alex Riolexus Ario.

**Visualization:** Maureen Katusiime.

**Writing – original draft:** Maureen Katusiime, Gerald Rukundo, Lilian Bulage.

**Writing – review & editing:** Maureen Katusiime, Steven Ndugwa Kabwama, Benon Kwesiga, Damian Rutazaana, Alex Riolexus Ario, Julie Harris.

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
