## [Decision Letter · Decision Letter 0]

12 Oct 2021

PGPH-D-21-00604

Malaria Outbreak Facilitated by Engagement in Activities near Swamps Following Increased Rainfall and Limited Preventive Measures: Oyam District, Uganda.

Dear Dr. Katusiime,

Thank you for submitting your manuscript to PLOS Global Public Health. After careful consideration, we feel that it has merit but does not fully meet PLOS Global Public Health’s publication criteria as it currently stands. Therefore, we invite you to submit a revised version of the manuscript that addresses the points raised during the review process.

We look forward to receiving your revised manuscript.

Kind regards,

Ruth Ashton, Ph.D.

Academic Editor

Journal Requirements:

1. In the Methods, please clarify that participants provided oral consent. Please also state in the Methods:

- Why written consent could not be obtained

- How oral consent was documented

For more information, please see our guidelines for human subjects research: https://journals.plos.org/globalpublichealth/s/submission-guidelines#loc-human-subjects-research

3. Please provide separate figure files in .tif or .eps format only, and remove any figures embedded in your manuscript file.  If you are using LaTeX, you do not need to remove embedded figures.

4. We have noticed that you have uploaded supporting information but you have not included a list of legends.  Please add a full list of legends for all supporting information files (including figures, table and data files) after the references list. 

5. Since your data is not available for proprietary reasons, please explain via email why the data is not available. Please also include the contact information for the third party organization that should be contacted should other researchers want to request access to this data and please include the full citation of where the data can be found. We also request that you verify with us via email that any researcher will be able to obtain the data set in the same manner that the you have obtained it. If you feel you are unwilling or unable to adhere to this policy, please explain your reasons by return email and your exemption request will be escalated to the editor for approval. Your exemption request will be handled independently and will not hold up the peer review process, but will need to be resolved should your manuscript be accepted for publication. One of the Editorial team will be in touch if they require more information.

6. Please provide us with a direct link to the base layer of the map used in figure 2 and figure 4 and ensure this location is also included in the figure legend. 

Please note that, because all PLOS articles are published under a CC BY license (creativecommons.org/licenses/by/4.0/), we cannot publish proprietary maps such as Google Maps, Mapquest or other copyrighted maps. If your map was obtained from a copyrighted source please amend the figure so that the base map used is from an openly available source.

Please note that only the following CC BY licences are compatible with PLOS licence: CC BY 4.0, CC BY 2.0  and CC BY 3.0, meanwhile such licences as CC BY-ND 3.0 and others are not compatible due to additional restrictions. If you are unsure whether you can use a map or not, please do reach out and we will be able to help you. 

The following websites are good examples of where you can source open access or public domain maps:

Additional Editor Comments (if provided):

Reviewers' comments:

Reviewer's Responses to Questions

**Comments to the Author**

1. Does this manuscript meet PLOS Global Public Health’s publication criteria? Is the manuscript technically sound, and do the data support the conclusions? The manuscript must describe methodologically and ethically rigorous research with conclusions that are appropriately drawn based on the data presented.

Reviewer #1: Partly

Reviewer #2: No

2. Has the statistical analysis been performed appropriately and rigorously?

Reviewer #1: No

Reviewer #2: Yes

3. Have the authors made all data underlying the findings in their manuscript fully available (please refer to the Data Availability Statement at the start of the manuscript PDF file)?

Reviewer #1: Yes

Reviewer #2: Yes

4. Is the manuscript presented in an intelligible fashion and written in standard English?

Reviewer #1: Yes

Reviewer #2: No

5. Review Comments to the Author

Reviewer #1: Malaria outbreak investigation is an important topic in global public health especially in endemic countries.

The authors did a case control study to investigate the scope of the outbreak, identified exposures associated with increased

transmission, and recommended evidence-based control measures. However the paper does not flow especially in the methods section. The authors also did entomological monitoring but missed out important parameters such as sporozoite rates and entomological inoculation rates. The mosquito house resting density and mosquito age parameters measured were not interpreted. For example If 65% of the mosquitoes collected were blood fed, what this mean on bed net use?

More comments below:

Abstract

Line 29-30 in the statement “We conducted entomological and environmental assessments.”. Elaborate why this was done like you did in line 30-31 in the statement “We interviewed 83 case-patients and 83 asymptomatic age- and parish-matched controls to determine exposures associated with illness”

Change breeding sites to Anopheles aquatic habitats

For the statement “sand pits (25/350mls) and brick pits (16/350mls) had the highest average larval concentrations” which is not clear. If you aimed at quantifying larval densities, you would measure the number of larvae per scoop.

Methods:

You need to further describe the study area including economic activities in the area.

You need not clarify the health service delivery in the sub-county ; How many health units are in the sub-county, the levels of these health units, how many were visited.

The two statements from line 95-98 sound the same.

In line 120, is 2km a radius? This normally done using transects of given width and length.

You will need to define all aquatic habitats (what you call breeding sites).

I did not see how rainfall data was obtained.

In line 126, change the word trapped to collected

Inline 130, name the arerosol used.

In line 131, give a reference of the identification key used

Petri dishes should be changed to petri dishes (From upper case)

In line 133, you mean mosquitoes were dissected, ovaries dried before examining them.

Entomological assessment should be broken done to Adult mosquito collection and Mosquito larval survey.

Line 152-154, the word “stagnant water”, see my comment on breeding site.

You mixed environmental assessment and larval survey (Make the method flow logically)

Reviewer #2: There are several references provided that are rather strange or unavailable for instance, 2 and 4, or even repeated under a different listing (for instance, 4 and 37 appear to be the same). Please consider updating the citation using a proper reference manager

Lines 69-70: It’s not clear what timepoint the problems discussed are referring to. They all seem to be from well before the present study duration, as indicated by the references provided. Please consider providing more relevant evidence for these or more current potential justifications for the outbreak.

• For instance, it’s unlikely inappropriate case management was the major driver, given the current high testing rates – Please also provide a definition for in/appropriate case management.

• It may also not be weak surveillance, given the existence of what’s known as integrated disease surveillance and response (IDSR) – Please consider reviewing IDSR and supporting your argument accordingly.

Line 73 – Here you confirm the effectiveness of surveillance with indication that districts were able to detect the upsurge and reported it. Is this not conflicting with your earlier assertions?

Line 80 – you suggest that you are going to recommend evidence-based control measures. However, this upsurge was in 2019, would this be appropriate for 2021?

Line # 84-85: It is not entirely clear what you meant by the “wet season” that lasts 11 months of the year. Please provide some clarification on this

Line #86: States that the “dry season is hot and mostly cloudy”, which part of Uganda would this be whose dry season takes this description?

Altogether, it appears like the calendar year is comprised of 13 months, which is rather confusing (11 dry and 2 wet!?)

Lines 84-89: You describe 4 seasons including: wet, dry, hot, and cool. Please simplify or make clearer which is which?

Line 94-95 – It’s not clear what you mean by “increasing specificity associated with cases” were you assessing specificity?

Line 97 – Among the data sets you were collecting was “date of onset of clinical symptoms”. Would you please provide some indication of availability and completeness of this particular data variable in HMIS?

Line 114 – It’s not clear whether the assumption here is that village and parish population estimates remained the same between 2014 when the census was conducted and 2019 when data for this study was collected. Why were population estimates of 2014 used in 2019?

Line 119 – Please clarify what you mean by “physical features” are these hills, valleys, buildings, …?

Line 120 – What approach was used to determine the 2Km distance that was used?

Line 165-166 – It’s not clear what the sample of 20 confirmed case-persons were interviewed to do?

This paragraph seems to jump from one thing to another and leaves the reader wondering what it’s leading up to

Line 178-179 – Given your definition of exposure as being based on fever, it’s not clear how you determined controls that were exposed. Please clarify

Line 179 – Sounds confusing that you measured “radius of a river”. How did you do this?

Lines 198-199 – Given no mention of rainfall data till this point, how did you use the t-test to determine rainfall differences between years?

Line 202: Please consider using a more suitable title, it is not clear what you mean by “descriptive epidemiology”

Figure 1 has no key, making it hard to interpret

Figure 2 has some hidden parts (large parts of it are black), it’s not clear what’s being portrayed

In Figure 3, what do the numbers on the x-axis represent? Also, given that week of symptom onset was recorded per individual among sampled cases, it’s not clear where the big numbers of cases on the y-axis come from

Case Control section

Table 1: With 1:1 matching, how come you had 4 cases in the under 5 years of age and only 1 among controls in this group? Then also, 30% cases from the 5-17 years group and 37% among controls?

Additionally, it’s not clear what units the average net usage per household and that per week take on. Are these days or what?

Table 2: Please revise this table to include the actual p-values and include the reference group for each variable indicated

Line 270 indicates engaging in activities, which activities? Also, the statement is not supported by your results given that those <100m were not shown to be associated, at least not significantly.

Line 307-309 – It’s not clear from your results that the activities within swamps increased or were different from year to year and therefore, these statements may not be supported by your results.

Line 315 – For inappropriate case management referred to, the evidence used is well before the duration of this study i.e. 2004, 2009 and 2012. This has certainly changed over the years and so please consider providing more relevant evidence for your claim.

Lines 326-332 – This section seems to be fit for the background of your paper. It’s not clear which results are being discussed in this section. Please consider revising or moving to a more appropriate location.

Line 341-342 – It’s not clear what this sentence is presenting. Simply because the report indicated (that is also not available at the reference provided) highlights the possible effects of weak surveillance doesn’t make an appropriate source of evidence of the existence of weak surveillance.

Line 348-349 – The relevance of this sentence is also not clear. There seems to be a mix up of monitoring that was apparently done well from your statements and response from higher levels (district and MoH) which appears to have been delayed. Please make this paragraph a little more focused.

Lines 358-359 – It is surprising that your first limitation is about recording of deaths. This study you indicated was looking at confirmed malaria cases. At what point did the deaths then become a source of limitations in studying cases?

Line 370 – The statement “to respond timely after …” is rather grammatically confusing, please consider revising.

6. PLOS authors have the option to publish the peer review history of their article (what does this mean?). If published, this will include your full peer review and any attached files.

**Do you want your identity to be public for this peer review?** For information about this choice, including consent withdrawal, please see our Privacy Policy.

Reviewer #1: **Yes: **Musiime K. Alex

Reviewer #2: No

---

## [Decision Letter · Decision Letter 1]

6 May 2022

PGPH-D-21-00604R1

Malaria Outbreak Facilitated by Engagement in Activities near Swamps Following Increased Rainfall and Limited Preventive Measures: Oyam District, Uganda.

Dear Dr. Katusiime,

Thank you for submitting your manuscript to PLOS Global Public Health. After careful consideration, we feel that it has merit but does not fully meet PLOS Global Public Health’s publication criteria as it currently stands. Therefore, we invite you to submit a revised version of the manuscript that addresses the points raised during the review process.

Please note that while the reviewers are satisfied with the changes made, they have a few additional comments that they feel would further improve the manuscript. 

We look forward to receiving your revised manuscript.

Kind regards,

Ruth Ashton, Ph.D.

Academic Editor

Journal Requirements:

Additional Editor Comments (if provided):

Reviewers' comments:

Reviewer's Responses to Questions

**Comments to the Author**

1. If the authors have adequately addressed your comments raised in a previous round of review and you feel that this manuscript is now acceptable for publication, you may indicate that here to bypass the “Comments to the Author” section, enter your conflict of interest statement in the “Confidential to Editor” section, and submit your "Accept" recommendation.

Reviewer #1: (No Response)

Reviewer #2: All comments have been addressed

2. Does this manuscript meet PLOS Global Public Health’s publication criteria? Is the manuscript technically sound, and do the data support the conclusions? The manuscript must describe methodologically and ethically rigorous research with conclusions that are appropriately drawn based on the data presented.

Reviewer #1: Yes

Reviewer #2: Yes

3. Has the statistical analysis been performed appropriately and rigorously?

Reviewer #1: Yes

Reviewer #2: Yes

4. Have the authors made all data underlying the findings in their manuscript fully available (please refer to the Data Availability Statement at the start of the manuscript PDF file)?

Reviewer #1: Yes

Reviewer #2: Yes

5. Is the manuscript presented in an intelligible fashion and written in standard English?

Reviewer #1: Yes

Reviewer #2: Yes

6. Review Comments to the Author

Reviewer #1: I thank the reviewers for this improved version of the manuscript and addressing my initial comments. However, on reading through this version, I suggested minor revisions. I believe, if addressed, they would greatly improve the manuscript.

Abstract:

In line 30, I suggest remove vector behavior, since in your methods and results, it’s not captured.

I think “to determine vector density and potential aquatic Anopheles habitats” clear states what you did.

In line 43, “scoop+”, do you mean positive for larvae?

In line 47, use conservative statement like, “This outbreak was likely facilitated by Anopheles aquatic habitats near homes’’

From line 178-183, seems to state that all the aquatic habitats sampled were containers.

Why do you need household index and container index? In addition, you can’t say houses are positive for larvae because we know larvae are found in aquatic habitats.

To compute larval density, I would simply calculate average larvae per dip and express it as average larvae per dip per habitat type. The aim would be to know which habitat type is most productive for Anopheles. If you did not record number of dips per habitat during data collection, then it’s difficult, you could record the number of larvae per habitat type although this depends on sampling effort.

In line 268, “We found a parity rate of 14% and longevity of 2.1 days”, You mean on average mosquitoes live for 2 days. Line 275-276 need to be changed if you agree with my comments on calculating larval density.

In line 313, the statement “Notably, 20% of mosquitoes captured in homes were male, potentially representing a very high larval density” may not be accurate. It implies male mosquitoes are an indicator of larval density, yet in the methods you calculated larval density.

Correct the error in line 331 “itis”

In line 335, see my comment on container and house index.

Reviewer #2: These are just a few quick revisions to be addressed

Line # 69-70: The sentence is not quite clear

Line # 72: It's not clear here. Do you mean good data quality, completeness, and ability to analyze data compromise the adequacy of malaria surveillance system? This would be misleading to the reader.

Line # 91: How is it that you conducted an outbreak? This is rather confusing

Line # 319-320: The sentence is not quite clear and in a couple other places this occurs where words are unclear because there's no space between what appears to be the right separate words.

Line # 384-385: The sentence that starts with "A decline in cases..." is not so clear

Line # 400: It's not at all clear what your recommendation of "mass case management ..." means in real life

7. PLOS authors have the option to publish the peer review history of their article (what does this mean?). If published, this will include your full peer review and any attached files.

**Do you want your identity to be public for this peer review?** For information about this choice, including consent withdrawal, please see our Privacy Policy.

Reviewer #1: **Yes: **Alex K. Musiime

Reviewer #2: No

---

## [Editor Report · Decision Letter 2]

13 Jul 2022

Malaria Outbreak Facilitated by Engagement in Activities near Swamps Following Increased Rainfall and Limited Preventive Measures: Oyam District, Uganda.

PGPH-D-21-00604R2

Dear Ms Katusiime,

We are pleased to inform you that your manuscript 'Malaria Outbreak Facilitated by Engagement in Activities near Swamps Following Increased Rainfall and Limited Preventive Measures: Oyam District, Uganda.' has been provisionally accepted for publication in PLOS Global Public Health.

Best regards,

Ruth Ashton, Ph.D.

Academic Editor